# Baseline clinical features of COVID-19 patients, delay of hospital admission and clinical outcome: A complex relationship

**Cédric Dananché**[1,2]*, **Christelle Elias**[1,2], **Laetitia Hénaff**[2], **Sélilah Amour**[1], **Elisabetta Kuczewski**[1], **Marie-Paule Gustin**[2], **Vanessa Escuret**[3,4], **Mitra Saadatian-Elahi**[1,2], **Philippe Vanhems**[1,2]

1 Service Hygiène, Epidémiologie, Infectiovigilance et Prévention, Hospices Civils de Lyon, Lyon, France, 2 Team « Public Health, Epidemiology and Evolutionnary Ecology of Infectious Diseases (PHE3ID) », Centre International de Recherche en Infectiologie, Institut National de la Santé et de la Recherche Médicale U1111, Centre National de la Recherche Scientifique Unité Mixte de Recherche 5308, École Nationale Supérieure de Lyon, Université Claude Bernard Lyon 1, Lyon, France, 3 Laboratoire de Virologie, Institut des Agents Infectieux, Hospices Civils de Lyon, Lyon, France, 4 VirPath, Centre International de Recherche en Infectiologie, Institut National de la Santé et de la Recherche Médicale U1111, Centre National de la Recherche Scientifique Unité Mixte de Recherche 5308, École Nationale Supérieure de Lyon, Université Claude Bernard Lyon 1, Lyon, France

* cedric.dananche@chu-lyon.fr

**Data Availability Statement:** All relevant data are within the manuscript and its Supporting information files.

## Abstract

### Introduction

Delay between symptom onset and access to care is essential to prevent clinical worsening for different infectious diseases. For COVID-19, this delay might be associated with the clinical prognosis, but also with the different characteristics of patients. The objective was to describe characteristics and symptoms of community-acquired (CA) COVID-19 patients at hospital admission according to the delay between symptom onset and hospital admission, and to identify determinants associated with delay of admission.

### Methods

The present work was based on prospective NOSO-COR cohort data, and restricted to patients with laboratory confirmed CA SARS-CoV-2 infection admitted to Lyon hospitals between February 8 and June 30, 2020. Long delay of hospital admission was defined as ≥6 days between symptom onset and hospital admission. Determinants of the delay between symptom onset and hospital admission were identified by univariate and multiple logistic regression analysis.

### Results

Data from 827 patients were analysed. Patients with a long delay between symptom onset and hospital admission were younger (p<0.01), had higher body mass index (p<0.01), and were more frequently admitted to intensive care unit (p<0.01). Their plasma levels of C-reactive protein were also significantly higher (p<0.01). The crude in-hospital fatality rate was lower in this group (13.3% *versus* 27.6%), p<0.01. Multiple analysis with correction for multiple testing showed that age ≥75 years was associated with a short delay between symptom

**Funding:** PV received partial funding by REACTing (Research and ACTion targeting emerging infectious diseases), Institut national de la santé et de la recherche médicale (INSERM), France and Fondation AnBer (http://fondationanber.fr), France. The study funders had no role in study design, data collection and analysis, decision to publish, or preparation of the manuscript. There was no other additional external funding received for this study.

**Competing interests:** The authors have read the journal's policy and have the following competing interests: PV received grants and fees from Anios, Pfizer, Astellas, MSD, Gilead and Sanofi. These grants and fees were not related to this present manuscript. This does not alter our adherence to PLOS ONE policies on sharing data and materials. There are no patents, products in development or marketed products associated with this research to declare. All other authors have no competing interest to declare.

onset and hospital admission (≤5 days) (aOR: 0.47 95% CI (0.34–0.66)) and CRP>100 mg/L at admission was associated with a long delay (aOR: 1.84 95% CI (1.32–2.55)).

## Discussion

Delay between symptom onset and hospital admission is a major issue regarding prognosis of COVID-19 but can be related to multiple factors such as individual characteristics, organization of care and severe pathogenic processes. Age seems to play a key role in the delay of access to care and the disease prognosis.

## Introduction

Severe acute respiratory syndrome coronavirus 2 (SARS-CoV-2), the pathogenic agent of coronavirus disease 2019 (COVID-19), is a new coronavirus that emerged from China in December 2019 [1]. The virus causes infections of the lower or upper respiratory tract of varying severity, from the common cold to severe pneumonia, respiratory failure, and death [2]. Male sex, old age, obesity, biological parameters such as biomarkers of inflammation, are known to be associated with severe disease [3–6]. The presence of at least one comorbidity (e.g. cardiovascular or chronic pulmonary disease) has been reported in 60% to 90% of hospitalized COVID-19 patients [7]. Regarding access to care, a positive association between a long time interval between COVID-19 diagnosis or hospital admission and the occurrence of severe disease or death has been described but little data are available [8, 9].

The NOSO-COR study is an international observational, prospective, multicentric study carried out in 13 French hospitals and hospitals affiliated with the GABRIEL network [10] in order to estimate the prevalence and incidence of SARS-CoV-2 infection and to assess the associated characteristics among healthcare workers and patients. Our previous results, based on 417 patients, showed that in COVID-19 patients, age as well as delay between symptom onset and hospital admission were both associated with ICU admission [11].

In order to investigate this observation further in a larger number of patients, this work aimed #1) to describe in detail the characteristics and symptoms of community-acquired (CA) COVID-19 patients at hospital admission according to the delay between symptom onset and hospital admission, #2) to identify determinants associated with the delay of admission.

## Methods

The present work was restricted to patients with CA SARS-CoV-2 infection admitted to Lyon University Hospitals (Hospices Civils de Lyon, HCL), between February 8 and June 30, 2020 with complete data at discharge or death. Compared to our previously published work [11], this analysis was based on the same cohort, but with a larger number of patients (n = 827 versus n = 417 in the previously published paper).

The detailed protocol of the NOSO-COR study is available online [10]. Briefly, any patient who presented an infectious syndrome based on the WHO definition of COVID-19 as of March 30, 2020 [12], and was hospitalized for a period of at least 24 hours, was included. Patients with positive real-time Reverse Transcriptase–Polymerase Chain Reaction (RT-PCR) results were defined as laboratory confirmed SARS-CoV-2 infections. A CA SARS-CoV-2 infection was defined as a patient with symptom onset before or at hospital admission.

Demographic characteristics, underlying comorbidities, clinical and biological parameters and patient outcome data were collected. The clinical outcomes were monitored up to hospital discharge or in-hospital death.

Delay between symptom onset and hospital admission was computed as the difference between the two dates. Data higher than the 99th percentile of the distribution of the delay were considered as outliers and removed. We categorized the delay in 2 groups according to the median of its distribution: "Short delay" when occurrence of symptoms was 5 days (included) or less before hospital admission; "long delay" when occurrence of symptoms was ≥6 days before hospital admission. This choice was guided by the distribution of the variable (S1 Fig and S1 Table). Admission to ICU included patients directly admitted to ICU and those hospitalized in a medical ward and subsequently transferred to ICU during their hospitalization. Continuous variables were reported as median and interquartile range (IQR) with comparisons using the Mann-Whitney U test. Qualitative variables were computed as number of individuals (n) and frequency (%) using the $\chi2$ or Fisher exact test as appropriate for comparison. The trend of the delay between symptom onset and hospital admission by age group and the trend of the proportion of patients with ICU hospitalization by age group were assessed using Cuzick's test. All tests were 2-tailed, with $p < 0.05$ considered statistically significant.

The logistic regression analysis was performed with delay between symptom onset and hospital admission as the dependent variable. Explanatory variables were first tested by univariate regression. Interaction of each variable with the variable age ≥ 75 years and sex was tested 1 by 1. Variables with $p < 0.10$ in univariate analysis were added in a multivariate logistic regression model (i.e. the complete model). Then, using a backward selection technique, variables were removed one by one from the complete model, in order to keep the simplest model to predict the delay (i.e. the parsimonious model). Holmes correction for multiple testing was applied to the final parsimonious model. Statistical analysis was performed using STATA 13® (College Station, TX, USA).

### Ethics

The study was approved by the clinical research and ethics committee of Ile-de-France V on March 8, 2020 (NOSO-COR, ClinicalTrials: NCT04290780). This study is an observational study based on patient medical records, and all data were fully anonymized at the time of collection. According to French law, patients or parents/guardians of minors received written information on this observational study, and their non-opposition to the use of their data was obtained.

### Results

Between February 8 and June 30, 2020, a total of 1,150 patients hospitalized in Lyon University Hospital were included. Of the 905 (78.7%) patients with a CA SARS-CoV-2 infection, 67 (7.4%) were excluded because of an unknown delay between symptom onset and hospital admission, and 11 (1.2%) were considered as outliers, leaving 827 patients for the final analysis.

The characteristics of the overall study population, according to the delay between symptom onset and hospital admission, are detailed in Table 1. The median age was 73 years (IQR: 61–84), 55.9% of patients were male, 21.9% were admitted to intensive care unit (ICU) or transferred to ICU during their hospitalization. Cardiovascular disease was the most frequent comorbidity (53.2%). Patients with a longer delay were younger ($p < 0.01$); had a higher body mass index ($p < 0.01$); were more frequently admitted to ICU ($p < 0.01$). They presented different characteristics and symptoms at admission more frequently, particularly cough ($p < 0.01$),

**Table 1. Characteristics of the study population according to delay between symptom onset and hospital admission, Lyon University Hospital (NOSO-COR Study), February 8–June 30, 2020.**

| Characteristics | All patients n = 827 | Patient with a short delay between symptom onset and hospital admission (≤5 days): n = 421 | Patient with a long delay between symptom onset and hospital admission (≥6 days): n = 406 | p-value |
|---|---|---|---|---|
| Male gender, n (%) | 462 (55.9) | 219 (52.0) | 243 (59.9) | 0.02 |
| Age, median (IQR) | 73 (61–84) | 79 (65–87) | 69 (54–78) | <0.01 |
| Age ≥ 75 y., n (%) | 384 (46.4) | 246 (58.4) | 138 (34.0) | <0.01 |
| BMI, median (IQR) | 26.2 (23.1–29.7) [664] | 25.7 (22.5–29.2) [349] | 26.7 (23.7–30.0) [315] | <0.01 |
| BMI ≥ 30, n (%) | 153 (23.0) | 76 (21.8) | 77 (24.4) | 0.42 |
| Comorbidities, n (%) | | | | |
| *Cardiovascular disease* | 440 (53.2) | 256 (60.8) | 184 (45.3) | <0.01 |
| *Diabetes* | 191 (23.1) | 106 (25.2) | 85 (20.9) | 0.15 |
| *Malignancy* | 146 (17.7) | 92 (21.9) | 54 (13.3) | 0.01 |
| *Chronic kidney disease* | 115 (13.9) | 72 (17.1) | 43 (10.6) | 0.07 |
| *Chronic lung disease* | 103 (12.5) | 60 (14.3) | 43 (10.6) | 0.11 |
| *Chronic liver disease* | 55 (7.5) [736] | 28 (7.1) [395] | 27 (7.9) [341] | 0.67 |
| *Immunodeficiency* | 48 (5.8) | 27 (6.4) | 21 (5.2) | 0.45 |
| Smoking status, n (%) | [626] | [304] | [322] | 0.30 |
| *Current smoker* | 29 (4.6) | 18 (5.9) | 11 (3.4) | |
| *Ex-smoker* | 197 (31.5) | 97 (31.9) | 100 (31.1) | |
| *Never smoker* | 400 (63.9) | 189 (62.2) | 211 (65.5) | |
| *Characteristics* | *All patients n = 827* | *Patient with a short delay between symptom onset and hospital admission (≤5 days): n = 421* | *Patient with a long delay between symptom onset and hospital admission (≥6 days): n = 406* | *p-value* |
| *Temperature at admission, median (IQR)* | 38.0 (37.1–38.6) [739] | 38.0 (37.2–38.5) [383] | 38.0 (37.1–38.8) [356] | 0.12 |
| *Symptoms at admission, n (%)* | | | | |
| *History of fever/chills* | 674 (81.5) | 330 (78.4) | 344 (84.7) | 0.02 |
| *Weakness* | 584 (70.6) | 282 (67.0) | 302 (74.4) | 0.02 |
| *Cough* | 560 (67.7) | 252 (59.9) | 308 (75.9) | <0.01 |
| *Shortness of breath* | 550 (66.5) | 257 (61.1) | 293 (72.2) | 0.01 |
| *Diarrhea* | 229 (27.8) | 103 (24.5) | 126 (31.0) | 0.04 |
| *Pain* | 225 (27.2) | 97 (23.0) | 128 (31.5) | <0.01 |
| *Myalgia* | 144 (17.4) | 60 (14.3) | 84 (20.7) | 0.02 |
| *Abdominal pain* | 62 (7.5) | 36 (8.6) | 26 (6.4) | 0.24 |
| *Chest pain* | 50 (6.0) | 19 (4.5) | 31 (7.6) | 0.06 |
| *Joint pain* | 12 (1.5) | 5 (1.2) | 7 (1.7) | 0.99 |
| *Nausea* | 107 (12.9) | 50 (11.9) | 57 (14.0) | 0.35 |
| *Headache* | 105 (12.7) | 43 (10.2) | 62 (15.3) | 0.03 |
| *Confusion* | 93 (11.3) | 63 (14.5) | 30 (7.4) | 0.01 |
| *Runny nose* | 72 (8.7) | 33 (7.8) | 39 (9.6) | 0.37 |
| *Ageusia* | 64 (7.7) | 14 (3.3) | 50 (12.3) | <0.01 |
| *Anosmia* | 58 (7.0) | 12 (2.9) | 46 (11.3) | <0.01 |
| *Sore throat* | 34 (4.1) | 16 (3.8) | 18 (4.4) | 0.65 |
| *Biological parameters, median (IQR)* | | | | |
| *White blood cells (G/L)* | 6.36 (4.87–8.59) [769] | 6.46 (4.71–8.62) [394] | 6.26 (4.95–8.52) [375] | 0.76 |
| *Neutrophils (G/L)* | 4.75 (3.28–6.83) [768] | 4.73 (3.19–7.07) [393] | 4.82 (3.35–6.61) [375] | 0.69 |

*(Continued)*

**Table 1.** (Continued)

| | | | | |
|---|---|---|---|---|
| *Lymphocytes (G/L)* | 0.95 (0.64–1.31) [767] | 0.95 (0.63–1.32) [392] | 0.94 (0.64–1.30) [375] | 0.78 |
| *CRP (mg/L)* | 71.4 (30.0–135.1) [711] | 65.0 (27.3–116.7) [369] | 86.6 (32.4–142.2) [342] | <0.01 |
| Admission to ICU, n (%) | 181 (21.9) | 76 (18.1) | 105 (25.9) | <0.01 |
| *Admission directly to ICU* | 136 (16.4) | 46 (10.9) | 90 (22.2) | <0.01 |
| *Admission to general ward and transfer to ICU during hospitalization* | 45 (5.4) | 30 (7.1) | 15 (3.6) | 0.03 |
| Death during hospitalization, n (%) | 170 (20.6) | 116 (27.6) | 54 (13.3) | <0.01 |

NOTE: in square brackets []: number of data available for the variable. If no square brackets, there is no missing data for the variable. BMI: Body mass index, CRP: C-reactive protein, IQR: Interquartile range. [e]reference category: white blood cells $\leq$ 10 G/L, [f]reference category: Neutrophils $\leq$ 7.5 G/L,. [g]reference category: Lymphocytes $\geq$ 1 G/L, [h]reference category: CRP $\leq$ 100 mg/L

weakness (p = 0.02), shortness of breath (p = 0.01), pain (p<0.01), ageusia (p<0.01) and anosmia (p<0.01). The frequency of reported confusion was lower in patients with a longer delay (p<0.01), whereas the plasmatic levels of C-reactive protein (CRP) were significantly higher (p<0.01) than in patients with a shorter delay. The crude in-hospital fatality rate was lower in the group of patients with a longer delay (13.3%) than in patients with a short delay (27.6%), p<0.01.

Median delay between symptom onset and hospital admission was 5 days (IQR: 3–9). This delay was significantly longer in ICU-hospitalized patients compared to those without ICU admission (7 days [IQR: 4–10] vs 5 days [IQR: 2–8], p<0.01).

As shown in Fig 1, the delay between symptom onset and hospital admission and the proportions of patients hospitalized to ICU decreased with age for patients above 60 years of age (p<0.01 and p<0.01, respectively). Patient characteristics according to age are displayed in S2 Table. The results showed that the median delay between symptom onset and hospital admission decreased in older patients (P<0.01). The proportion of ICU admission decreased in older patients (P<0.01), while death during hospitalization increased in this population (P<0.01). Overall, older patients had more comorbidities, such as cardiovascular diseases (P<0.01), malignancy (P<0.01) or chronic kidney diseases (P<0.01) compared to younger patients.

Overall, 711 patients (86.0%) for whom complete data were available, were included in the logistic regression analysis. Table 2 depicts the association between underlying comorbidities, clinical features and biological parameters at admission and the delay between symptom onset and hospitalization. In multivariate analysis, age $\geq$75 years (p<0.01), and confusion at admission (p = 0.02), were associated with a short delay between symptom onset and hospital admission. On the contrary, weakness (p = 0.02), cough (p = 0.01), ageusia (p = 0.02), anosmia (p = 0.03) and CRP>100 mg/L at admission (p<0.01) were associated with a long delay between symptom onset and hospital admission. After correction for multiple testing, only two variables remained significant in the model: age $\geq$75 years was associated with a short delay between symptom onset and hospital admission (p<0.01) and CRP>100 mg/L at admission was associated with a long delay between symptom onset and hospital admission (p<0.01).

## Discussion

The results of our study showed that during the course of SARS-CoV-2 infection, older patients ($\geq$75 years) present earlier to the hospital. An association between younger age and

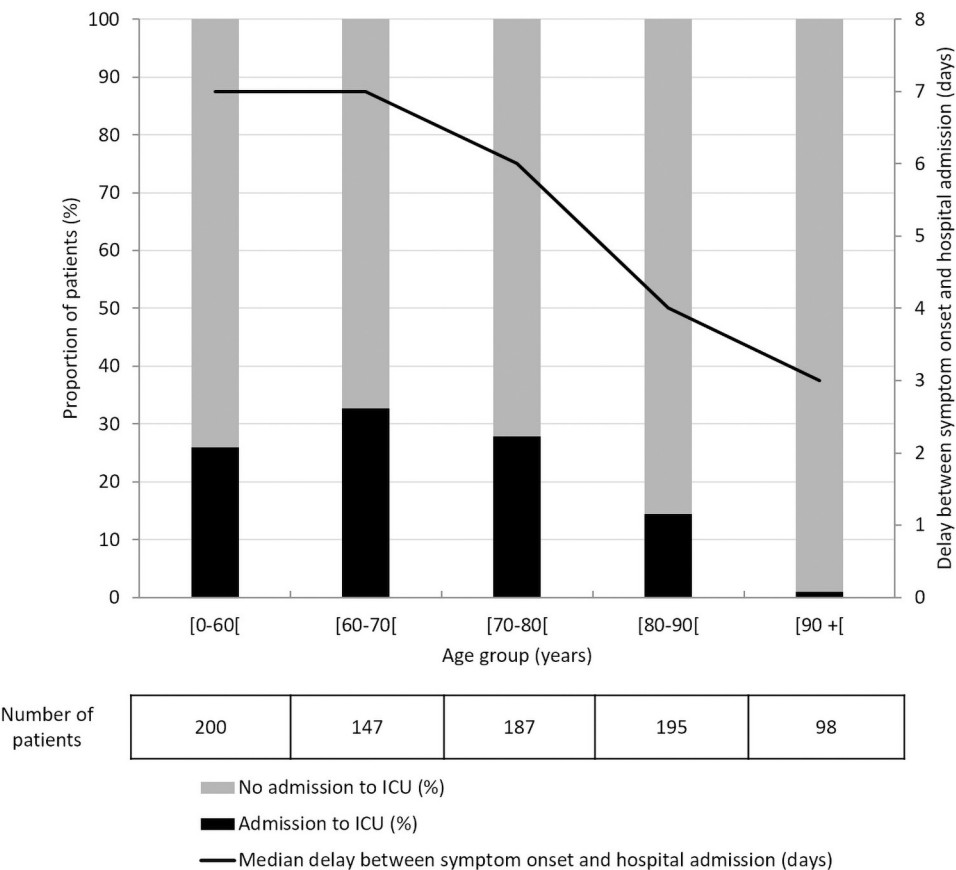

**Fig 1. Delays between symptom onset and hospital admission and proportions of admission to ICU according to patient age, Lyon University Hospital (NOSO-COR Study), February 8–June 30, 2020.** Note: Admission to ICU includes patients directly admitted to ICU and patients hospitalized in a medical ward and subsequently transferred to ICU during their hospitalization.

longer delay to hospital admission was also found in Brazil [13]. Higher prevalence of some infectious diseases (e.g. bacteremia) together with more severe presentation of other infectious diseases such as influenza in the elderly could explain, at least partly, the observed shorter delay between symptom onset and hospital admission in this population [14]. Aging plays a role in the pathogenic process and is also an independent determinant of outcome [11, 13]. However, age could also be a determinant of behavior and perception of risk and severity. A recent review of the literature showed that risk perception increases with age [15]. Some researchers even postulate the existence of "a unique integrated compensatory biological/behavioral immune system" by reasoning that the weakened immune system in older adults could be compensated by more prudent behavior [16, 17]. Age would therefore play a key role in the delay of access to care and the prognosis of COVID-19. It suggests also that young people without comorbidities may neglect the first clinical signs of COVID-19 and the need to consult health care facilities. Indeed, due to the widespread information communicated by social media on the low COVID-19 morbidity and mortality in this population; adolescents and young adults may have a lower risk perception of COVID-19 for themselves compared to older people, thus delaying the seeking of care [18–20].

A recent study carried out in China showed a positive association between severity of COVID-19 and the interval between symptom onset and diagnosis [9]. Similarly, the low

**Table 2. Association between patient characteristics and delay between symptom onset and hospital admission, Lyon University Hospital (NOSO-COR Study), February 8–June 30, 2020.**

| Characteristics | Crude OR of the risk of long delay between symptom onset and hospital admission (>5 d) (CI 95%)[a] | p-value | Adjusted OR of the risk of long delay between symptom onset and hospital admission (>5 d) (ie. complete model) (CI 95%)[b] | p-value | Adjusted OR of the risk of long delay between symptom onset and hospital admission (>5 d) (ie. parsimonious model) (CI 95%)[c] | p-value | Corrected p-value (Holmes multiple testing correction) |
|---|---|---|---|---|---|---|---|
| Male gender | 1.31 (0.98–1.77) | 0.07 | 0.90 (0.65–1.26) | 0.55 | | | |
| Age ≥ 75 y | 0.36 (0.26–0.49) | <0.01 | 0.54 (0.38–0.77) | <0.01 | 0.47 (0.34–0.66) | <0.01 | <0.01 |
| BMI ≥ 30 | 1.17 (0.79–1.74) | 0.43 | | | | | |
| Comorbidities | | | | | | | |
| Cardiovascular disease | 0.53 (0.39–0.72) | <0.01 | 0.79 (0.56–1.11) | 0.18 | | | |
| Diabetes | 0.90 (0.63–1.27) | 0.55 | | | | | |
| Malignancy | 0.55 (0.37–0.82) | <0.01 | 0.66 (0.43–1.01) | 0.06 | 0.65 (0.42–0.99) | 0.05 | 0.08 |
| Chronic lung disease | 0.73 (0.47–1.14) | 0.16 | | | | | |
| Chronic kidney disease | 0.61 (0.39–0.94) | 0.03 | 0.93 (0.57–1.51) | 0.76 | | | |
| Chronic liver disease | 1.13 (0.63–2.03) | 0.69 | | | | | |
| Immunodeficiency | 0.79 (0.41–1.50) | 0.47 | | | | | |
| Smoking status | | | | | | | |
| Current smoking | 1 (ref.) | | | | | | |
| Ex smoker | 1.52 (0.67–3.47) | 0.32 | | | | | |
| Never smoking | 1.53 (0.69–3.40) | 0.29 | | | | | |
| Characteristics | Crude OR of the risk of long delay between symptom onset and hospital admission (>5 d) (CI 95%)[a] | p-value | Adjusted OR of the risk of long delay between symptom onset and hospital admission (>5 d) (ie. complete model) (CI 95%)[b] | p-value | Adjusted OR of the risk of long delay between symptom onset and hospital admission (>5 d) (ie. parsimonious model) (CI 95%)[c] | p-value | Corrected p-value (Holmes multiple testing correction) |
| Symptoms at admission | | | | | | | |
| History of fever/chills | 1.57 (1.07–2.30) | 0.02 | 1.28 (0.83–1.95) | 0.26 | | | |
| Weakness | 1.40 (1.01–1.95) | 0.04 | 1.52 (1.06–2.18) | 0.02 | 1.53 (1.07–2.19) | 0.02 | 0.08 |
| Cough | 2.13 (1.54–2.96) | <0.01 | 1.47 (1.03–2.10) | 0.03 | 1.56 (1.10–2.22) | 0.01 | 0.08 |
| Shortness of breath | 1.51 (1.10–2.06) | 0.01 | 1.20 (0.84–1.70) | 0.31 | | | |
| Diarrhea | 1.45 (1.04–2.01) | 0.03 | 1.22 (0.85–1.75) | 0.28 | | | |
| Myalgia | 1.83 (1.24–2.69) | <0.01 | 1.38 (0.89–2.13) | 0.15 | | | |
| Abdominal pain | 0.83 (0.48–1.43) | 0.50 | | | | | |
| Chest pain | 1.60 (0.86–2.98) | 0.14 | | | | | |
| Joint pain | 1.52 (0.48–4.84) | 0.48 | | | | | |
| Nausea | 1.15 (0.75–1.75) | 0.53 | | | | | |
| Headache | 1.66 (1.07–2.57) | 0.02 | 1.02 (0.62–1.69) | 0.94 | | | |
| Confusion | 0.41 (0.25–0.68) | <0.01 | 0.56 (0.33–0.96) | 0.04 | 0.52 (0.31–0.89) | 0.02 | 0.08 |
| Runny nose | 1.39 (0.83–2.34) | 0.22 | | | | | |
| Ageusia | 5.06 (2.57–9.94) | <0.01 | 2.58 (1.17–5.71) | 0.02 | 2.61 (1.20–5.71) | 0.02 | 0.08 |
| Anosmia | 5.03 (2.48–10.19) | <0.01 | 2.28 (0.98–5.29) | 0.06 | 2.50 (1.10–5.70) | 0.03 | 0.08 |
| Sore throat | 1.08 (0.51–2.30) | 0.84 | | | | | |
| Biological parameters | | | | | | | |

(Continued)

**Table 2.** (Continued)

| | | | | | | | |
|---|---|---|---|---|---|---|---|
| White blood cells > 10 G/L | 0.86 (0.57–1.30) | 0.48 | | | | | |
| Neutrophils > 7.5 G/L | 0.88 (0.60–1.31) | 0.54 | | | | | |
| Lymphocytes < 1 G/L | 1.10 (0.82–1.48) | 0.51 | | | | | |
| CRP > 100 mg/L | 1.72 (1.27–2.33) | <0.01 | 1.76 (1.24–2.48) | <0.01 | 1.84 (1.32–2.55) | <0.01 | <0.01 |

NOTE: BMI: Body mass index, CRP: C-reactive protein

[a] in univariate analysis,

[b] in multivariate analysis, the complete model included all variables with p<0.10 in univariate analysis,

[c] in multivariate analysis, the model included the variables retained after backward selection, ie. age ≥75 y. o., malignancy, weakness, cough, confusion, ageusia, anosmia and CRP > 100 mg/L

fatality rate from COVID-19 in South Korea was attributed to a rapid presentation to health care facilities as soon as symptoms appeared [21]. This observation has also been reported for other viral respiratory diseases, such as influenza, where more severe clinical presentation with admission to ICU have been associated to late laboratory diagnosis [22]. This work showed that a long delay between symptom onset and hospital admission was associated with an increase in ICU admission. However, it showed also that patient characteristics (e.g. age, comorbidities such as cardiovascular diseases) changed according to the delay between symptom onset and hospital admission, and could act as confounding factors. Indeed, the differences in patient characteristics, biological signs or imaging procedures, in particular age, inflammatory levels, or computed tomography detected lung lesions according to the delay between symptom onset and hospital admission have been reported to be closely linked to the clinical course of the disease and the prognosis [23, 24]. Investigating the association between delay of hospital admission and ICU admission is complex in the context of our study because of the heterogeneity of the population, the close relationship between ICU admission, age and patient health conditions. In addition, it is known that the first wave of COVID-19 created overloads in the healthcare system due to the limited number of beds in ICU [25, 26]. Our data did not allow to investigate whether the lower proportion of ICU admission in older patients was the result of or the presence of a large proportion of patients with do-not-resuscitate decisions.

Plasmatic level of CRP>100 mg/L was found to be associated with a long delay between symptom onset and hospital admission. It is known that hyper inflammation can occur during the clinical course of the disease [27] and that elevated plasmatic level of inflammatory biomarkers (such as CRP) were associated with COVID-19 severity [28]. Our results showed that clinical presentation at admission could be associated with the delay between symptom onset and hospital admission, even if the variables did not reach significance after correction for multiple testing (p = 0.08). Confusion at admission tended to be associated with a short delay between symptom onset and hospital admission, as already reported in the literature [29]. This observation could be explained by an early COVID-19 diagnosis and hospital admission in cognitively impaired persons, for example institutionalized persons or individuals with dementia [30]. Ageusia and anosmia tended to be associated with a long delay between symptom onset and hospital admission, whilst they are typically early symptoms [31]. A classification bias might explain this observation, as these two symptoms have a long time of recovery, up to several weeks [31].

None of the studied comorbidities was associated with the delay between symptom onset and hospital admission, despite a statistical significance of cardiovascular diseases, malignancy, chronic kidney diseases in univariate analysis. These factors, described as risk factors for COVID-19 severity in other studies [32–34] are closely associated with age, which seems to be the major determinant of the delay between symptom onset and hospital admission.

The prospective character of the study design and data collection using a standardized protocol are the main strengths of the study implemented early in the pandemic. The study could bring additional findings that will inspire future exploration of the complex link between patient characteristics, delay of hospital admission and outcome for this emerging infection.

This study has however some limitations. We did not collect parameters such as socioeconomic status, known to be associated with access to care for other diseases [21]. Also, a bias might exist as governmental guidelines changed from March 2020 and during the entire COVID-19 pandemic, the French government recommended the general population to first call their general practitioner (GP) in case of suspicion of COVID-19. GPs had to prescribe nasopharyngeal swabs for COVID-19 confirmation, to assess the clinical severity of the COVID-19 confirmed patients and to call the emergency department to inform the hospital for those requiring hospitalization. Finally, the statistical power of the study could be too low to detect an independent effect between delay of hospital admission and patient characteristics.

In conclusion, delay between symptom onset and hospital admission is a key issue regarding prognosis. It can be related to individual characteristics, organization of care and/or a pathogenic process increasing the need for healthcare (e.g. exacerbation of a chronic disease). The respective contribution of these factors remains challenging to determine. Improving access to diagnosis, clinical surveillance and access to care after symptom onset is essential to avoid severe clinical symptoms in patients with high risk of severe disease. It could also reduce the rate of transmission during the interval between symptom onset and hospital admission, particularly in patients with a delayed diagnosis (e.g. young patients), by informing them about the prevention measures to apply. In this context, patient surveillance using a telemedicine system could be an innovative approach [35].

## Supporting information

**S1 Fig. Distribution of the variable "delay between symptom onset and hospital admission".**
(DOCX)

**S1 Table. Characteristics of the study population according to the quartiles of delay between symptom onset and hospital admission, Lyon University Hospital (NOSO-COR Study), February 8–June 30, 2020.**
(DOCX)

**S2 Table. Characteristics of the study population according to age group, Lyon University Hospital (NOSO-COR Study), February 8–June 30, 2020.**
(DOCX)

**S1 File. Minimal underlying dataset.**
(XLSX)

## Acknowledgments

The authors express their gratitude to: COVID-Outcomes-HCL Consortium (affiliation for all: Hospices Civils de Lyon): Laurent Argaud, Frédéric Aubrun, Marc Bonnefoy, Maude

Bouscambert-Duchamp, Roland Chapurlat, Dominique Chassard, Christian Chidiac, Michel Chuzeville, Cyrille Confavreux, Sébastien Couraud, Gilles Devouassoux, Isabelle Durieu, Michel Fessy, Sylvain Gaujard, Alexandre Gaymard, Arnaud Hot, Bruno Lina, Géraldine Martin Gaujard, Emmanuel Morelon, Vincent Piriou, Véronique Potinet, Jean-Christophe Richard, Thomas Rimmelé, Pascal Sève, Alain Sigal, Karim Tazarourte. 2) the Department of Health Data of Lyon Hospital: A. Duclos, F. Gueyffier, S. Vautier and M. Hervé for the creation and management of e-CRF, 3) Clinical research associates for data collection and data entry: V. Artizzu, L. Bissuel, S. Bennina, L. Dehina-Khenniche, A. Darrin, M. Grange, B. Robin, 4) staff of the virology laboratory of the Lyon hospital: Claire Bandolo, Genevieve Billaud, Maude Bouscambert-Duchamp, Emilie Frobert, Alexandre Gaymard, Laurence Josset, Christophe Ramiere, Isabelle Schuffenecker, Solange Telusson, Martine Valette, Florence Morfin for providing the results of RT-PCR tests.

The authors also thank Michelle Grange for editing the manuscript.

## Author Contributions

**Conceptualization:** Philippe Vanhems.

**Data curation:** Christelle Elias, Sélilah Amour, Elisabetta Kuczewski, Mitra Saadatian-Elahi.

**Formal analysis:** Cédric Dananché, Marie-Paule Gustin.

**Funding acquisition:** Philippe Vanhems.

**Investigation:** Laetitia Hénaff, Sélilah Amour, Elisabetta Kuczewski, Vanessa Escuret.

**Methodology:** Cédric Dananché, Christelle Elias, Laetitia Hénaff, Vanessa Escuret, Mitra Saadatian-Elahi, Philippe Vanhems.

**Project administration:** Laetitia Hénaff, Mitra Saadatian-Elahi, Philippe Vanhems.

**Resources:** Mitra Saadatian-Elahi, Philippe Vanhems.

**Supervision:** Mitra Saadatian-Elahi, Philippe Vanhems.

**Validation:** Cédric Dananché, Christelle Elias, Laetitia Hénaff, Marie-Paule Gustin.

**Visualization:** Cédric Dananché, Christelle Elias, Sélilah Amour, Marie-Paule Gustin, Mitra Saadatian-Elahi, Philippe Vanhems.

**Writing – original draft:** Cédric Dananché, Mitra Saadatian-Elahi, Philippe Vanhems.

**Writing – review & editing:** Cédric Dananché, Christelle Elias, Laetitia Hénaff, Sélilah Amour, Elisabetta Kuczewski, Marie-Paule Gustin, Vanessa Escuret, Mitra Saadatian-Elahi, Philippe Vanhems.

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
