## [Decision Letter · Decision Letter 0]

10 Sep 2021

PONE-D-21-19254Baseline clinical features of COVID-19 patients, delay of hospital admission and clinical outcome: a complex relationshipPLOS ONE

Dear Dr. Dananché,

Thank you for submitting your manuscript to PLOS ONE. After careful consideration, we feel that it has merit but does not fully meet PLOS ONE’s publication criteria as it currently stands. Therefore, we invite you to submit a revised version of the manuscript that addresses the points raised during the review process.

We look forward to receiving your revised manuscript.

Kind regards,

Itamar Ashkenazi

Academic Editor

PLOS ONE

Journal Requirements:

 This work was partially supported by REACTing (Research and ACTion targeting emerging infectious diseases), Institut national de la santé et de la recherche médicale (INSERM), France and Fondation AnBer (http://fondationanber.fr), France. 

 This work was partially supported by REACTing (Research and ACTion targeting emerging infectious diseases), Institut national de la santé et de la recherche médicale (INSERM), France and Fondation AnBer (http://fondationanber.fr), France. 

6. Please amend either the abstract on the online submission form (via Edit Submission) or the abstract in the manuscript so that they are identical.

Reviewers' comments:

Reviewer's Responses to Questions

**Comments to the Author**

1. Is the manuscript technically sound, and do the data support the conclusions?

Reviewer #1: Yes

Reviewer #2: Partly

Reviewer #3: Yes

2. Has the statistical analysis been performed appropriately and rigorously? 

Reviewer #1: Yes

Reviewer #2: No

Reviewer #3: No

3. Have the authors made all data underlying the findings in their manuscript fully available?

Reviewer #1: Yes

Reviewer #2: Yes

Reviewer #3: Yes

4. Is the manuscript presented in an intelligible fashion and written in standard English?

Reviewer #1: Yes

Reviewer #2: Yes

Reviewer #3: Yes

5. Review Comments to the Author

Reviewer #1: This is interesting study, however, I have several important comments.

Major comment)

1. The standard value of the day, 5 day, is reliable? Actually, that day is the mean value of them in this study. If authors divide them to 1Q, 2-3Q, and 4Q (quartile), then they might have statistically significant results. To define proper standard value of the day (the definition of the 'deyal'), authors need to confirm the distribution of the day (spot distribution figure?).

2. Some specific clinical characteristics (age, gender, symptoms...) can induce delay of the admision, as authors said. However, I think admission to ICU is not attributable factor to delay of the admission, but it can be the results of the dalay of the admission. Then, the data of the ICU admission can be descrbied in the bottom of the table. In addition, in multiple analysis, that variable should be deleted.

3. Authors can find significant factors (including, age, delay of the admission...) to induce admission to ICU (this might be the outcome variables).

Minor comments)

1. the name of the group can be modified as general form (such as 'group 1 and group 2' or 'control and delayed group')

2. In table 1. please describe the symbol of the [ ]

Reviewer #2: This paper focused on the delay of hospital admission in COVID-19 and stated that age appeared to be a key determinant of it. The delay of hospital admission is of great interest, since it might affect the clinical outcome as the authors pointed out.

I have several comments:

1. The authors should separate the cause of delay from its effect on the clinical outcome. In the present analysis, ICU admission was adopted as an explanatory variable for delay, but it could not be the determinant of delay since ICU admission occurred after the admission. Authors should exclude the ICU admission from the variables to assess the determinant of delay. Plus, in order to assess the impact of delay on the clinical consequence, outcome (ICU admission or death) should be the dependent variable and delay should be one of the explanatory variables along with other clinical characteristics and biomarkers.

2. The authors pointed out in figure 1 that the proportion of patients with ICU admission decreased with age for patients above 60 years old. Is this because older patients’ condition was milder? Or any other reason other than severity, such as increased proportion of patients with the do-not-resuscitate (DNR) decisions? If the latter is the case, caution should be needed in discussing about clinical outcome based on ICU admission.

3. As authors pointed out, government guideline for the hospital visits presumably affected the interval between onset and hospital visit. I believe it would be beneficial to incorporate the data concerning the government guideline (e.g., whether the admission of a given patient was before or after the change of guideline).

4. Authors should confirm whether the reference is appropriate for a given statement. For example, reference [2] and [3] is a “clinical practice” article, which might be inappropriate in L59–L60, L61–L62, respectively.

5. In L64–L66, authors stated previous report was questionable. The reason should be clarified.

6. The statement in L109–L110 “Overall, 711 patients (86.0%) for whom complete data were available” should be in the Result section.

7. Renaming the groups into more easy ones e.g., “Long delay” / “Short delay” instead of Gr#1/Gr#2 might be more reader friendly.

8. Decimal places should be consistent for p values in the tables.

9. In L176, authors stated that incidence of infectious diseases increases dramatically with age, but it’s not true. It’s variable according to kind of infectious diseases.

10. In the X axis of Figure 1, using en dash (e.g., 0–60) might be better label for descripting age rage.

Reviewer #3: The manuscript studies the association between the delay between symptom-onset date and hospital-admission date and clinical and demographic factors in 827 COVID-19 patients. The patients were divided into two groups, based on the delay in their admission to hospital, with the long (short) delay group having a delay higher (lower) than the median (which is 6 days). In the abstract, it is stated that factors associated with the delay are identified by means of multivariate logistic regression. In fact, univariate logistic regression and comparisons of factors between groups were also performed.

The main results are based on the multivariate logistic regression, which shows that age, confusion at admission, and subsequent transfer to ICU were positively associated with a short delay while weakness, cough, ageusia, anosmia, and CRP>100 mg/L at admission were negatively associated (although their statistical significance was not properly addressed, see below).

I recommend the publication of this manuscript provided that the following issues are addressed.

-The abstract reports a comparison of the characteristics of the two groups and provides p-values. It is not clear which statistical tests were actually used and this should be explained in details. The statement in the main-text methods section it is too vague as it only states "Mann-Whitney U test and Chi-square or Fisher exact test were used when appropriate and that trends were assessed using Cuzick's test or Spearman's rank correlation coefficient". Which test was used exactly for each comparison?

-It appears that a correction for multiple testing was not performed in the multivariable regression, while that should be included, especially given that many p-values are borderline (too close to the standard threshold at 0.05) and some feature may be strongly correlated. Performing a multiple-testing correction will show whether the association found in this study are statistically significant.

-I was surprised to see that, In table 2, the adjusted ORs of many features (e.g., comorbidities) for the multivariable regression are not included, while the crude ORs are always included. As the multivariable regression is a better tool than the univariate regression to study the associations, all adjusted ORs must be included.

minor comment:

- It is not clear to me what "prospectively collected" means in the following context:

"Demographic characteristics, underlying comorbidities, clinical and biological parameters and patient outcome data were collected prospectively" and I would appreciate a clear definition of its meaning.

6. PLOS authors have the option to publish the peer review history of their article (what does this mean?). If published, this will include your full peer review and any attached files.

Reviewer #1: No

Reviewer #2: **Yes: **Hiroaki Sasaki

Reviewer #3: No

---

## [Author Response · Author response to Decision Letter 0]

25 Oct 2021

October 25, 2021

Answer letter

Reference : PONE-D-21-19254.

R: Thank you, we have modified the article as requested.

R We have added the requested information in the ethics statement of the revised manuscript. 

 R: We have amended the statement as requested. Please see the revised manuscript. 

R: We have amended the statement as requested. Please see the revised manuscript. 

 This work was partially supported by REACTing (Research and ACTion targeting emerging infectious diseases), Institut national de la santé et de la recherche médicale (INSERM), France and Fondation AnBer (http://fondationanber.fr), France. 

R: We have added the above-mentioned statement in the revised manuscript. 

R: As requested, we have added the minimal data set as a Supporting Information file (S1 File).

6. Please amend either the abstract on the online submission form (via Edit Submission) or the abstract in the manuscript so that they are identical.

R: We have modified the abstracts so that they are now identical.

 Reviewer #1: This is interesting study; however, I have several important comments.

Major comments:

1. The standard value of the day, 5 day, is reliable? Actually, that day is the mean value of them in this study. If authors divide them to 1Q, 2-3Q, and 4Q (quartile), then they might have statistically significant results. To define proper standard value of the day (the definition of the 'deyal'), authors need to confirm the distribution of the day (spot distribution figure?).

R: Thank you for this comment. The value of 5 days used to define short delay and long delay is in fact the median value of the distribution, and not the mean value. This value appears to be more relevant than the mean value as the distribution of the delays is not a normal distribution. Using the median value of the delay is useful to divide the population into 2 groups of the same size.

The parameters of the distribution are the following: number of included patients n=827, median delay between symptom onset and hospital admission = 5 days, Q1=3 days, Q3=9 days, mean delay between symptom onset and hospital admission = 6.06 days, standard deviation = 4.56 days, Shapiro-Wilk test for normality: p<0.01.

The graph of the distribution of patients according to the delay between symptom onset and hospital admission is reported in Figure S1 of the revised manuscript. 

We described also the data in 4 groups according to the quartiles of the distribution (Q1-median-Q3), as suggested. Please see Table S1 in the revised version. No major differences were observed compared to the description with 2 groups, with a threshold at the median value. We concluded that using the median value of the distribution as a threshold was suitable for the description of our dataset and our analyses.

2. Some specific clinical characteristics (age, gender, symptoms...) can induce delay of the admission, as authors said. However, I think admission to ICU is not attributable factor to delay of the admission, but it can be the results of the delay of the admission. Then, the data of the ICU admission can be described in the bottom of the table. In addition, in multiple analysis, that variable should be deleted.

R: Indeed, ICU admission occurred after or at hospital admission.

The objective of the addition of this variable in the model was to determine if ICU admission was associated with the delay between symptom onset and hospital admission; not as an explanatory variable, but as a consequence of the delay.

However, we understand the comment of the reviewer and as suggested, we reported this variable at the bottom of the table and removed it from the multiple regression model.

3. Authors can find significant factors (including, age, delay of the admission...) to induce admission to ICU (this might be the outcome variables).

R: Our previously published paper (Vanhems P, Gustin M-P, Elias C, Henaff L, Dananché C, Grisi B, et al. Factors associated with admission to intensive care units in COVID-19 patients in Lyon-France. PLoS One. 2021;16: e0243709. doi:10.1371/journal.pone.0243709) aimed to study the associations between different factors and ICU admission. In this paper, ICU admission was the dependent variable and the delay between symptom onset and hospital admission, one of the explanatory variables. The results showed that the delay between symptom onset and hospital admission was associated with ICU admission.

This observation motivated us to better characterize the determinants of the delay between symptom onset and hospital admission.

According to the above-mentioned publication, and even with a larger population size (n=827 versus n=417 in the previously published paper), we did choose to not present a model with ICU admission as the outcome variable.

Minor comments:

1. the name of the group can be modified as general form (such as 'group 1 and group 2' or 'control and delayed group')

R: As suggested, we have changed the name of the groups to facilitate reading.

2. In table 1. please describe the symbol of the [ ]

R: We have specified the meaning of the symbol in the Table legend.

Reviewer #2: 

This paper focused on the delay of hospital admission in COVID-19 and stated that age appeared to be a key determinant of it. The delay of hospital admission is of great interest, since it might affect the clinical outcome as the authors pointed out.

I have several comments:

1. The authors should separate the cause of delay from its effect on the clinical outcome. In the present analysis, ICU admission was adopted as an explanatory variable for delay, but it could not be the determinant of delay since ICU admission occurred after the admission. Authors should exclude the ICU admission from the variables to assess the determinant of delay. Plus, in order to assess the impact of delay on the clinical consequence, outcome (ICU admission or death) should be the dependent variable and delay should be one of the explanatory variables along with other clinical characteristics and biomarkers.

R: Thank you for this comment. Indeed, ICU admission occurred after or at hospital admission. 

The objective of the addition of this variable in the model was to determine if ICU admission was associated with the delay between symptom onset and hospital admission; not as an explanatory variable, but as a consequence of the delay.

However, we understand the comment of the reviewer and removed the variable ICU admission of the multiple regression analysis.

Our previously published paper (Vanhems P, Gustin M-P, Elias C, Henaff L, Dananché C, Grisi B, et al. Factors associated with admission to intensive care units in COVID-19 patients in Lyon-France. PLoS One. 2021;16: e0243709. doi:10.1371/journal.pone.0243709) aimed to study the associations between different factors and ICU admission. In this paper, ICU admission was the dependent variable and the delay between symptom onset and hospital admission, one of the explanatory variables. The results showed that the delay between symptom onset and hospital admission was associated with ICU admission.

This observation motivated us to better characterize the determinants of the delay between symptom onset and hospital admission.

According to the above-mentioned publication, and even with a larger population size (n=827 versus n=417 in the previously published paper), we did choose to not present a model with ICU admission as the outcome variable.

2. The authors pointed out in figure 1 that the proportion of patients with ICU admission decreased with age for patients above 60 years old. Is this because older patients’ condition was milder? Or any other reason other than severity, such as increased proportion of patients with the do-not-resuscitate (DNR) decisions? If the latter is the case, caution should be needed in discussing about clinical outcome based on ICU admission.

R: We compared patient characteristics according to age group in S2 Table of the revised manuscript. We also provided description of the results in the Results section of the revised manuscript lines 164-168.

Investigating the association between delay of hospital admission and ICU admission is complex in the context of our study because of the heterogeneity of the population, the close relationship between ICU admission, age and patient health conditions. In addition, it is known that the first wave of COVID-19 created overloads in the healthcare system due to the limited number of beds in ICU. Our data did not allow to investigate whether the lower proportion of ICU admission in older patients was the result of or the presence of a large proportion of patients with do-not-resuscitate decisions. 

We have added these details regarding limitations in the Discussion section lines 238-245.

3. As authors pointed out, government guideline for the hospital visits presumably affected the interval between onset and hospital visit. I believe it would be beneficial to incorporate the data concerning the government guideline (e.g., whether the admission of a given patient was before or after the change of guideline).

A: Indeed, governmental guidelines were changed from March 2020 and were in effect during the duration of the pandemic. The French government recommended the general population to first call their general practitioner (GP) in case of suspicion of COVID-19. GPs had to prescribe nasopharyngeal swabs for COVID-19 confirmation, to assess the clinical severity of the COVID-19 confirmed patients and to call the emergency department to inform the hospital for those requiring hospitalization. We have added these details to the Discussion section lines 272-278.

4. Authors should confirm whether the reference is appropriate for a given statement. For example, reference [2] and [3] is a “clinical practice” article, which might be inappropriate in L59–L60, L61–L62, respectively.

R: We have replaced these 2 references by more appropriate articles:

Zhou F, Yu T, Du R, Fan G, Liu Y, Liu Z, Xiang J, Wang Y, Song B, Gu X, Guan L, Wei Y, Li H, Wu X, Xu J, Tu S, Zhang Y, Chen H, Cao B. Clinical course and risk factors for mortality of adult inpatients with COVID-19 in Wuhan, China: a retrospective cohort study. Lancet. 2020 Mar 28;395(10229):1054-1062. doi: 10.1016/S0140-6736(20)30566-3. Epub 2020 Mar 11. 

Gao YD, Ding M, Dong X, Zhang JJ, Kursat Azkur A, Azkur D, Gan H, Sun YL, Fu W, Li W, Liang HL, Cao YY, Yan Q, Cao C, Gao HY, Brüggen MC, van de Veen W, Sokolowska M, Akdis M, Akdis CA. Risk factors for severe and critically ill COVID-19 patients: A review. Allergy. 2021 Feb;76(2):428-455. doi: 10.1111/all.14657. Epub 2020 Dec 4. PMID: 33185910

5. In L64–L66, authors stated previous report was questionable. The reason should be clarified.

R: We have reworded the sentence: “Regarding access to care, a positive association between a long time interval between COVID-19 diagnosis or hospital admission and the occurrence of severe disease or death has been described but little data are available”. Please see the revised manuscript lines 64-66.

6. The statement in L109–L110 “Overall, 711 patients (86.0%) for whom complete data were available” should be in the Result section.

R: As suggested, this statement has been included in the Results section. Please see the revised manuscript lines 178-179.

7. Renaming the groups into more easy ones e.g., “Long delay” / “Short delay” instead of Gr#1/Gr#2 might be more reader friendly.

R: We have changed the name of the groups to facilitate reading.

8. Decimal places should be consistent for p values in the tables.

R: We have harmonized the number of decimals for p-values in the revised version of the manuscript.

9. In L176, authors stated that incidence of infectious diseases increases dramatically with age, but it’s not true. It’s variable according to kind of infectious diseases.

R: We agree with the reviewer and we have changed the sentence. Please see the revised manuscript lines 209-212.

10. In the X axis of Figure 1, using en dash (e.g., 0–60) might be better label for descripting age range.

R: Thank you, we have corrected the Figure 1 accordingly.

Reviewer #3: The manuscript studies the association between the delay between symptom-onset date and hospital-admission date and clinical and demographic factors in 827 COVID-19 patients. The patients were divided into two groups, based on the delay in their admission to hospital, with the long (short) delay group having a delay higher (lower) than the median (which is 6 days). In the abstract, it is stated that factors associated with the delay are identified by means of multivariate logistic regression. In fact, univariate logistic regression and comparisons of factors between groups were also performed.

R: We have changed the Abstract to mention: “Determinants of the delay between symptom onset and hospital admission were identified by univariate and multiple logistic regression”.

The main results are based on the multivariate logistic regression, which shows that age, confusion at admission, and subsequent transfer to ICU were positively associated with a short delay while weakness, cough, ageusia, anosmia, and CRP>100 mg/L at admission were negatively associated (although their statistical significance was not properly addressed, see below).

I recommend the publication of this manuscript provided that the following issues are addressed.

-The abstract reports a comparison of the characteristics of the two groups and provides p-values. It is not clear which statistical tests were actually used and this should be explained in details. The statement in the main-text methods section it is too vague as it only states "Mann-Whitney U test and Chi-square or Fisher exact test were used when appropriate and that trends were assessed using Cuzick's test or Spearman's rank correlation coefficient". Which test was used exactly for each comparison?

R: We provided more detail about the statistical analysis in the Methods section. Please see the revised manuscript lines 103-109: “Continuous variables were reported as median and interquartile range (IQR) with comparisons using the Mann-Whitney U test. Qualitative variables were computed as number of individuals (n) and frequency (%) using the χ2 or Fisher exact test as appropriate for comparison. The trend of the delay between symptom onset and hospital admission by age group and the trend of the proportion of patients with ICU hospitalization by age group were assessed using Cuzick’s test. All tests were 2-tailed, with p<0.05 considered statistically significant.” 

-It appears that a correction for multiple testing was not performed in the multivariable regression, while that should be included, especially given that many p-values are borderline (too close to the standard threshold at 0.05) and some feature may be strongly correlated. Performing a multiple-testing correction will show whether the association found in this study are statistically significant.

R: As suggested, in the final multiple regression model (ie. the parsimonious model), we performed a multiple-testing correction using Holmes correction. Please see the revised manuscript Table 2. Results showed that plasmatic level of CRP >100 mg/L was significantly associated with a longer delay between symptom onset and hospital admission (p<0.01). Conversely, old age (≥75 years) was significantly associated with a shorter delay between symptom onset and hospital admission (p<0.01).

Confusion at admission tended to be associated with a short delay between symptom onset and hospital admission; whereas weakness, cough, ageusia and anosmia at admission tended to be associated with a longer delay between symptom onset and hospital admission, despite a lack of significance (p=0.08 for each variable). Please see the revised manuscript lines 185-188. The Discussion section was also changed accordingly (lines 249-257).

-I was surprised to see that, In table 2, the adjusted ORs of many features (e.g., comorbidities) for the multivariable regression are not included, while the crude ORs are always included. As the multivariable regression is a better tool than the univariate regression to study the associations, all adjusted ORs must be included.

R: We have rewritten the Methods section in order to clarify the analyses. Please see the revised manuscript lines 110-118: “The logistic regression analysis was performed with delay between symptom onset and hospital admission as the dependent variable. Explanatory variables were first tested by univariate regression. Interaction of each variable with the variable age ≥ 75 years and sex was tested 1 by 1. Variables with p<0.10 in univariate analysis were added in a multivariate logistic regression model (i.e. the complete model). Then, using a backward selection technique, variables were removed one by one from the complete model, in order to keep the simplest model to predict the delay (i.e. the parsimonious model). Holmes correction for multiple testing was applied to the final parsimonious model. Statistical analysis was performed using STATA 13® (College Station, TX, USA).”

Moreover, in the Table 2, we mentioned all adjusted OR in the complete model and in the parsimonious model.

minor comment:

- It is not clear to me what "prospectively collected" means in the following context:

"Demographic characteristics, underlying comorbidities, clinical and biological parameters and patient outcome data were collected prospectively" and I would appreciate a clear definition of its meaning.

R: This study is a prospective study according to the definition found in TL Lash, TJ VanderWeele, S Haneuse, KJ Rothman. Wolters Kluwer, 2021. Modern Epidemiology, 4th edition, chapter 6 Study design, page 116: “When the person-time accumulates after the study begins, it is said to be a prospective study. In this situation, the exposure status is ordinarily recorded before disease occurrence, although there are exceptions.”. In our study, participants were enrolled as soon as they were diagnosed as infected with SARS-CoV-2. In the present analysis, they were included in the study at hospital admission; as the population was restricted to patients with community-acquired SARS-CoV-2 infections.

For more clarity, we have removed the word “prospectively”, as the Introduction section mentions that the NOSO-COR study is an observational, prospective study.

---

## [Decision Letter · Decision Letter 1]

15 Nov 2021

PONE-D-21-19254R1Baseline clinical features of COVID-19 patients, delay of hospital admission and clinical outcome: a complex relationshipPLOS ONE

Dear Dr. Dananché,

Thank you for submitting your manuscript to PLOS ONE. Following re-evaluation of your revised manuscript, the reviewers conveyed that most of the issues were clarified. Still, on minor issue remains. Therefore, we invite you to submit a revised version of the manuscript that addresses the point raised during the review process.

We look forward to receiving your revised manuscript.

Kind regards,

Itamar Ashkenazi

Academic Editor

PLOS ONE

Journal Requirements:

Reviewers' comments:

Reviewer's Responses to Questions

**Comments to the Author**

1. If the authors have adequately addressed your comments raised in a previous round of review and you feel that this manuscript is now acceptable for publication, you may indicate that here to bypass the “Comments to the Author” section, enter your conflict of interest statement in the “Confidential to Editor” section, and submit your "Accept" recommendation.

Reviewer #2: All comments have been addressed

Reviewer #3: All comments have been addressed

2. Is the manuscript technically sound, and do the data support the conclusions?

Reviewer #2: Yes

Reviewer #3: Partly

3. Has the statistical analysis been performed appropriately and rigorously? 

Reviewer #2: Yes

Reviewer #3: Yes

4. Have the authors made all data underlying the findings in their manuscript fully available?

Reviewer #2: Yes

Reviewer #3: Yes

5. Is the manuscript presented in an intelligible fashion and written in standard English?

Reviewer #2: Yes

Reviewer #3: Yes

6. Review Comments to the Author

Reviewer #2: (No Response)

Reviewer #3: The authors have addressed all of my previous concerns and clarified the details of the statistical analyses. I now also noticed that the sentence of line 78 "Our previous results showed that in COVID-19 patients, age as well as delay between symptom onset and hospital admission were both associated with ICU admission [11]" requires further clarifications. The authors should really include the number of patients in the cited study or state whether the cohort of reference [11] is the same as that of the submitted paper.

7. PLOS authors have the option to publish the peer review history of their article (what does this mean?). If published, this will include your full peer review and any attached files.

Reviewer #2: **Yes: **Hiroaki Sasaki

Reviewer #3: No

---

## [Author Response · Author response to Decision Letter 1]

16 Nov 2021

Reviewer #3: The authors have addressed all of my previous concerns and clarified the details of the statistical analyses. I now also noticed that the sentence of line 78 "Our previous results showed that in COVID-19 patients, age as well as delay between symptom onset and hospital admission were both associated with ICU admission [11]" requires further clarifications. The authors should really include the number of patients in the cited study or state whether the cohort of reference [11] is the same as that of the submitted paper.

R : Thank you for this comment. Our previous results were based on 417 patients. In the present work, we used the same cohort of reference, but with a larger number of patients. We have modified the Introduction section lines 70-71 and the Methods section line 84 accordingly.

---

## [Editor Report · Decision Letter 2]

2 Dec 2021

Baseline clinical features of COVID-19 patients, delay of hospital admission and clinical outcome: a complex relationship

PONE-D-21-19254R2

Dear Dr. Dananché,

We’re pleased to inform you that your manuscript has been judged scientifically suitable for publication and will be formally accepted for publication once it meets all outstanding technical requirements.

Kind regards,

Itamar Ashkenazi

Academic Editor

PLOS ONE

---

## [Editor Report · Acceptance letter]

31 Dec 2021

PONE-D-21-19254R2 

Baseline clinical features of COVID-19 patients, delay of hospital admission and clinical outcome: a complex relationship 

Dear Dr. Dananché:

I'm pleased to inform you that your manuscript has been deemed suitable for publication in PLOS ONE. Congratulations! Your manuscript is now with our production department. 

Kind regards, 

on behalf of

Dr. Itamar Ashkenazi 

Academic Editor

PLOS ONE